# Structural and Functional Validation of a Full-Thickness Self-Assembled Skin Equivalent for Disease Modeling

**DOI:** 10.3390/pharmaceutics14061211

**Published:** 2022-06-07

**Authors:** Bo Ram Mok, Su-Ji Shon, A Ram Kim, Carolyne Simard-Bisson, Israël Martel, Lucie Germain, Dong Hyun Kim, Jung U Shin

**Affiliations:** 1Department of Biomedical Science, CHA University, Seongnam 13488, Korea; mokbbo@gmail.com (B.R.M.); orange11026@naver.com (S.-J.S.); tmfdkdjssl@hanmail.net (A.R.K.); 2Centre de Recherche en Organogénèse Expérimentale de l’Université Laval/LOEX, and Department of Surgery, Faculty of Medicine, Université Laval, CHU de Québec-Université Laval Research Centre, Québec, QC G1J1Z4, Canada; carolyne.simard-bisson@crchudequebec.ulaval.ca (C.S.-B.); israel.martel@crchudequebec.ulaval.ca (I.M.); lucie.germain@fmed.ulaval.ca (L.G.); 3Department of Dermatology, CHA Bundang Medical Center, CHA University School of Medicine, Seongnam 13497, Korea; terios92@cha.ac.kr

**Keywords:** self-assembly tissue engineering, reconstructed skin, 3D skin, psoriasis model

## Abstract

Recently, various types of in vitro-reconstructed 3D skin models have been developed for drug testing and disease modeling. Herein, we structurally and functionally validated a self-assembled reconstructed skin equivalent (RSE) and developed an IL-17a-induced in vitro psoriasis-like model using a self-assembled RSE. The tissue engineering approach was used to construct the self-assembled RSE. The dermal layer was generated using fibroblasts secreting their own ECM, and the epidermal layer was reconstructed by seeding keratinocytes on the dermal layer. To generate the psoriatic model, IL-17A was added to the culture medium during the air–liquid interface culture period. Self-assembled RSE resulted in a fully differentiated epidermal layer, a well-established basement membrane, and dermal collagen deposition. In addition, self-assembled RSE was tested for 20 reference chemicals according to the Performance Standard of OECD TG439 and showed overall sensitivity, specificity, and accuracy of 100%, 90%, and 95%, respectively. The IL-17a-treated psoriatic RSE model exhibited psoriatic epidermal characteristics, such as epidermal hyperproliferation, parakeratosis, and increased expression of KRT6, KRT17, hBD2, and S100A9. Thus, our results suggest that a self-assembled RSE that structurally and functionally mimics the human skin has a great potential for testing various drugs or cosmetic ingredients and modeling inflammatory skin diseases.

## 1. Introduction

The development of physiologically relevant models of skin tissue for drug testing and disease modeling has been increasing. In addition, 3D reconstructed skin models aimed at replacing or supplementing animal models have been the focus of recent research efforts [1,2]. Ethical issues related to the use of animal models for skin toxicity and sensitivity testing, as well as the lack of accuracy of animal-to-human extrapolations, have also sparked further interest in this field [1,3,4].

Reconstructed human skin equivalents (RSEs) are widely used in skin research and in the cosmetic industry for purposes such as material screening, and various models have been developed accordingly. The in vitro 3D skin culture systems have attracted the attention of researchers as a means of reproducing physiological skin properties, including barrier function, 3D structure, and immune response, in a single system [5,6]. Following this trend, the OECD has provided guidelines for the use of in vitro 3D skin for research or as an alternative to animal testing. One of them pertains to in vitro testing methods for predicting skin irritation potential. The OECD recommends hazard tests for photo-toxicity and chemical-toxicity damage, to validate the ability of in vitro 3D skin equivalents, and suggests an in vitro procedure for it [7].

The OECD Guideline TG439 (Reconstructed Human Epidermis Test Method) provides a criterion for demonstrating the stability of the reconstructed epidermis against irritating chemicals (substances and mixtures). To date, seven reconstructed 3D human skin models have been officially registered with the OECD [4,7,8,9,10,11]; however, a self-assembled RSE model composed of human-derived dermis, epidermis, and extracellular matrix has not been registered. Therefore, in this study, we performed a test for toxicants using a self-assembled RSE model according to the TG439 guideline and investigated its ability to simulate native skin.

Psoriasis remains one of the most common and chronic cutaneous diseases associated with a Th17-mediated immune response [12]. Epidermal hyperplasia and abnormal differentiation are the main hallmarks of psoriasis [12,13]. To date, various in vitro monolayer models (2D models) and in vivo models have been widely used to study psoriasis [14,15]. The monolayer models of psoriasis have been widely used, despite the fact that they do not reflect the interactions between the different types of cells that constitute the human skin. Conversely, the in vivo models of psoriasis include mostly murine models, which are divided into spontaneous, genetically engineered, and imiquimod-cream-induced models; these models simulate the human skin better than the monolayer models [16,17]. However, murine models do not fully represent human skin diseases because of differences between species. For those reasons, in vitro-reconstructed 3D human skin equivalents have emerged as a research tool in this field. Although engineered 3D human skin equivalents using animal collagen have been used most commonly as an in vitro model of psoriasis, a self-assembled skin substitute may be a more physiologically relevant model because they contain human-derived cells exclusively, as well as an extracellular matrix [18,19]. In this study, we developed an in vitro psoriasis-like reconstructed human skin equivalent model using only human-derived cells and extracellular matrix and adding IL-17A.

We demonstrated that the self-assembled RSE recapitulates a histological morphology and structure of the native human epidermis and dermis without the use of exogenous materials. In addition, we also showed that the self-assembled RSE had excellent performance against the OECD TG439 guideline, and that it is possible to construct various disease models using this RSE. Therefore, we conclude that these models are useful not only for understanding the pathophysiology of skin diseases but also for the testing of the toxicity of cosmetic ingredients, biomaterials, and medications.

## 2. Materials and Methods

### 2.1. Cell Culture and Treatment

Human primary keratinocytes and dermal fibroblasts were freshly isolated from human skin samples. Skin donors provided written informed consent. Keratinocytes were maintained in a keratinocyte medium purchased from ATCC (Manassas, VA, USA), whereas dermal fibroblasts were cultured in Dulbecco’s Modified Eagle Medium (DMEM) containing 10% fetal bovine serum (Cytiva, Marlborough, MS, USA). Cells were cultured at 37 °C in a humid atmosphere containing 5% CO_2_.

### 2.2. Generation of the RSE

Reconstructed skins were prepared in quadruplicate as previously described [19]. Primary dermal fibroblasts were grown in DMEM containing 10% FBS with ascorbic acid for 4 weeks. During this period, the fibroblasts secreted their own extracellular matrix to form a dermal fibroblast sheet. Subsequently, the skin structure was constructed using two dermal fibroblast sheets. Briefly, primary keratinocytes were seeded onto a dermal sheet and cultured for 1 week to build a dermal–epidermal layered structure, and then one dermal–epidermal equivalent and one dermal sheet were stacked, to prepare a full-thickness skin layer, and were cultured in the air–liquid interface for 14 days.

To induce psoriasis-like inflammation, 100 ng/mL of a psoriatic cytokine, such as recombinant IL-17A (Peprotech, Cranbury, NJ, USA), was added to the culture medium.

### 2.3. Histological Analysis

The self-assembled RSE models were fixed in 4% paraformaldehyde and embedded in paraffin. In addition, 5-µm thick sections were stained with hematoxylin and eosin purchased from IHCWORLD (Woodstock, MD, USA) using standard procedures to visualize tissue morphology. Masson’s Trichrome staining was performed to identify collagen-rich regions by general protocol. Briefly, paraffin-embedded tissue sections were re-fixed in Bouin’s solution (IHCWORLD, Woodstock, MD, USA) to improve staining quality and then stained with hematoxylin and alanine blue (IHCWORLD, MD, USA) for identifying nuclei and collagen fibers, respectively. Histological examinations were performed using a digital camera (DP74, Olympus, Japan) coupled with an optical microscope (BX53, Olympus, Tokyo, Japan).

### 2.4. Scanning Electron Microscopy (SEM)

To examine the collagen bundle in the dermis, skin tissue samples were processed for SEM analysis. The tissues were rinsed with saline and fixed with 2.5% glutaraldehyde in 0.1 M sodium cacodylate buffer at 4 °C overnight, followed by fixation with 0.1 M sodium cacodylate buffer, 0.2 M sucrose, and 2 mM MgCl_2_. The fixed samples were then subjected to dehydration using an increasing gradient of ethanol. Then, the samples were critical point-dried and fractured to expose their cross sections. The samples were visualized using a Merlin, field emission scanning electron microscope (FS-SEM), operating at 10 kV (Zeiss, Oberkochen, Germany).

### 2.5. Transmission Electron Microscopy (TEM)

The basement membrane of the self-assembled RSE model was analyzed through TEM. In this experiment, skin samples were fixed with 2% glutaraldehyde and 0.1 M sodium cacodylate at 4 °C overnight, followed by fixation with 2% osmium tetroxide. The samples were then stained with 0.5% uranyl acetate for 30 min. After dehydration with an increasing gradient of ethanol and propylene oxide, infiltration with Spurr’s resin was performed overnight. The images were examined at 80 kV using a JEM-1011 electron microscope (JEOL, Tokyo, Japan).

### 2.6. Tissue Viability

Tissue viability was evaluated by performing a 3-(4,5-dimethylthiazol-2-yl)-2,5-diphenyl-2H-tetrazolium bromide (MTT) assay (Abcam, Cambridge, UK) according to the OECD guideline (Test Guideline No. 439, 2021). Briefly, the skin samples were exposed to the test substances (50 μL/cm^2^) for 30 min at room temperature and then washed with DPBS, followed by post-incubation with phenol-red free media at 37 °C for 42 h. After transferring the samples into the transwell, the samples were further incubated with 1 mL of MTT solution at 37 °C. After 3 h, the MTT solution was removed and the tissues were incubated with the solvent that was supplied by the manufacturer to extract the reduced formazan from the tissues at room temperature overnight. In addition, 100-μL extracted solutions were measured at 590 nm using a Multiskan FC Microplate Reader (Thermo Fisher Scientific, Waltham, MS, USA). DPBS and 5% SDS were used as a negative control and a positive control, respectively. Tissue viability was calculated relative to the negative control.

### 2.7. Immunostaining

The 4% formaldehyde-fixed and dehydrated skin tissue samples were embedded in paraffin wax and cut into 5-µm-thick sections. The skin sections were incubated with a primary antibody against human KRT10 (1:100; Abcam, Cambridge, UK), followed by incubation with the adequate secondary antibody. After staining, sections were counterstained with hematoxylin to provide contrast. For immunofluorescence staining, the skin tissues were quench-frozen and embedded in OCT. The cryosections (10 µm) were fixed in chilled acetone for 15 min and incubated overnight at 4 °C with anti-S100A9 (1:400; Abcam, Cambridge, UK), anti-HBD-2 (1:400; Abcam, Cambridge, UK), anti-Filaggrin (1:400; Abcam, Cambridge, UK), anti-Involucrin (1:400; Abcam, Cambridge, UK), anti-AO-1 (1:400; Abcam, Cambridge, UK), anti-CLDN-1 (1:400; Abcam, Cambridge, UK), anti-KRT6 (1:400; Abcam, Cambridge, UK), anti-KRT10 (1:400; Abcam, Cambridge, UK), and anti-KRT17 (1:400; Abcam, Cambridge, UK) primary antibodies, followed by Alexa Fluor 488- or 568-conjugated secondary antibodies and DAPI. The immunostained images were obtained using a digital camera (DP74; Olympus, Tokyo, Japan) coupled with an optical microscope (BX53; Olympus, Tokyo, Japan).

### 2.8. RNA Extraction and Quantitative Real-Time PCR

Total RNA from the 3D skin equivalents was extracted using Trizol reagent (Invitrogen, Crlsbad, CA, USA) according to the manufacturer’s instructions; then, 1000 ng RNA was reverse transcribed using M-MLV reverse transcriptase (Promega, Madison, WI, USA) and oligo dT primers. Subsequently, relative mRNA expression was performed by real-time PCR using the SYBR Green master mix (Bioneer, Daejeon, Korea), the synthesized cDNA, and self-designed primers. Primers were designed according to the mRNA sequence; the sequences that were employed for *S100A9* were 5′-CATGGAGGACCTGGACACAAA-3′ for forward sequence and 5′-CCCTCGTGCATCTTCTCGTG-3′ for the reverse sequence. The primer sequences for *DEFB4* were 5′-GCTTGATGTCCTCCCCAGACT-3′ for forward and 5′- CAGGATCGCCTATACCACCAAA-3′ for reverse. The primer sequences for *IL-1b* were 5′-GCACTACAGGCTCCGAGATGAA-3′ for forward and 5′-GTCGTTGCTTGGTTCTCCTTGT-3′ for reverse. The primer sequences for *MKi67* were 5′-CGTCCCAGTGGAAGAGTTGT-3′ for forward and 5′-CGACCCCGCTCCTTTTGATA-3′ for reverse. The sequences for *GAPDH*, a reference gene, were 5′- ACCACAGTCCATGCCATCAC-3′ for forward sequence and 5′-TCCACCACCCTGTTGCTGTA-3′ for reverse sequence. PCR conditions were: initial denaturation for 15 min at 95 °C; followed by 40 cycles of 95 °C for 15 s, 60 °C for 45 s and 72 °C for 30 s, for primer annealing and extension. Relative mRNA levels were calculated by normalization to the reference gene *GAPDH* using the 2^−ΔΔCT^ method.

### 2.9. Statistics

All statistical analyses were performed on raw data using GraphPad Prism 6.0 (GraphPad, La Jolla, CA, USA). Multiple comparison groups were analyzed with one-way ANOVA, followed by Tukey’s post-hoc test. All data are presented as mean ± SEM.

## 3. Results

### 3.1. Histological Morphology of the Self-Assembled RSE Model Was Similar to That of the Normal Human Skin

Before developing disease models, we generated self-assembled reconstructed human skin and evaluated its structure and function to verify its suitability for testing chemical irritation and the development of disease models. The self-assembled RSE model is a three-dimensional skin equivalent comprising fully human-derived epidermis and dermis. To produce it, we first prepared fibroblast sheets by allowing fibroblasts to secrete their own extracellular matrix in vitro for 4 weeks. Subsequently, keratinocytes were seeded and grown on top of the fibroblast sheet, and a full-thickness skin layer was prepared by stacking all these fibroblast sheets and a fibroblast–keratinocyte combined sheet. After 14 days of culture at the air–liquid interface, fully differentiated skin substitutes were generated without the use of exogenous extracellular matrix proteins or synthetic materials (Figure 1).

Tissue engineering methods generated full-thickness self-assembled RSE models. This model consists of fibroblasts and keratinocytes without exogenous ECM material. In this model, keratinocytes are seeded onto a dermal sheet containing autocrine ECM and stacked with the other dermal sheet to form a layer; then, it undergoes differentiation and cornification through air–liquid interface culture for 14 days to create a full-thickness skin layer.

The self-assembled RSE model exhibited a native human skin-like histological morphology consisting of a fully differentiated and multilayered epidermis and a dermis rich in ECM (Figure 2a–e). The thickness of the epidermal layer of the RSE was approximately 200 µm, comprised of the stratum basale (26.5 µm), stratum spinosum (118.3 µm), stratum granulosum (27.4 µm), and stratum corneum (27.6 µm) (Figure 2b); moreover, we confirmed that keratinocyte differentiation occurred normally with the development of each layer in the epidermal layer (Appendix A). Specifically, in the self-assembled RSE, the keratinocytes that constituted the basement membrane existed in a columnar shape and gradually changed to a flat shape as they moved upward (Appendix A). In addition, a well-organized basement membrane was observed. A TEM analysis revealed that the self-assembled RSE contained most of the components of the basement membrane, such as hemidesmosomes, a lamina lucida, and a lamina densa, as well as complex entangled keratin filaments (Figure 2c). Anchoring fibrils were also observed, suggesting that anchoring fibrils extending into the dermal layer would strengthen the bond between the epidermis and dermis, as in the human skin.

Keratin 10 (KRT10) is a member of the cytokeratin family that belongs to the superfamily of intermediate filament proteins. The major function of the keratin intermediate filament is to generate cell cohesion and prevent the acute rupture of the epithelial cell sheet under tension, suggesting that keratin proteins support the epidermal layer. The subtype of keratin expressed in each layer of the epidermis varies according to the differentiation of the keratinocytes, which is an indicator of normal epidermal development [20]. KRT10 is expressed in the suprabasal layer of the human skin [20], and the same expression pattern was observed in the self-assembled RSE, suggesting that the self-assembled RSE undergoes appropriate epidermal differentiation as the normal human skin (Figure 2d). In addition to the proper epidermal layer, the self-assembled RSE also demonstrated abundant collagen bundles in the dermal layer, which provided structural support for the RSE (Figure 2e).

Next, we examined the expression of barrier-associated proteins, such as filaggrin (FLG) and involucrin (IVL), and tight junction proteins, such as zonula occludens-1 (ZO-1) and claudin-1 (CLDN-1). According to previous reports, filaggrin and involucrin exist in the granular layer and contribute to the formation of the skin barrier [21,22]. Filaggrin is produced in the transition from granules to keratinous by dephosphorylation and cleavage of profilaggrin; it then forms a complex with the keratin filaments, leading to the formation of a fibrous matrix and strengthening the skin barrier [23]. Involucrin, expressed in the granular layer, contributes to the creation of a formidable skin barrier as a cross-linked component of the keratinocytes and the keratinized stratum corneum [4]. Tight junction proteins are expressed in the epidermal layer, mainly in the stratum granulosome. Tight junctions and the stratum corneum have a synergistic effect on the formation of a strong skin barrier [24]. In the 3D skin equivalents, ZO-1 and CLDN-1 were expressed throughout the epidermal layer [25]. Similarly, our results by immunofluorescence staining analysis showed the presence of relevant barrier proteins as well as tight junction proteins. FLG and IVL were expressed in the granular layer (Figure 2f,g); moreover, the expression of ZO-1 and CLDN-1 was confirmed in all layers of the RSE model (Figure 2h,i).

Collectively, these results revealed that the self-assembled RSE recapitulates a histological morphology and structure of the native human epidermis and dermis without the use of exogenous materials.

### 3.2. Biological Reactivity of the Self-Assembled RSE Model

After confirming the proper organization of the epidermis and dermis, we investigated the biological function of the self-assembled RSE by measuring tissue viability using 1% Triton X-100, SDS, and 20 reference chemicals according to the Performance Standard of OECD TG439. First, we performed an MTT assay and assessed the 50% effective time (ET_50_), which is defined as the time required for 1% Triton X-100 to reduce cell viability to 50%. In addition, the half maximal inhibitory concentration (IC_50_) was measured after performing the MTT assay using SDS. IC_50_ describes the concentration of SDS that is necessary to reduce cell viability to 50%. The ET_50_ of the self-assembled RSE in the presence of 1% Triton X-100 was 7 h and 45 min, and the IC_50_ was 0.9873 mg/mL in the presence of SDS (Figure 3a,b). Next, the ability of the self-assembled RSE to screen chemical toxicity according to the OECD TG439 guideline was evaluated using 20 reference chemicals, 10 non-irritants, and 10 irritants (Table 1). In this test, all RSEs should have higher tissue viability than positive controls and lower viability than negative controls. OECD standards also recommend that tissue viability should be 50–100% for nonirritating chemicals and less than 50% for irritating chemicals. Specificity was evaluated by relative tissue viability of nonirritating chemicals, sensitivity was evaluated by relative tissue viability of irritating chemicals, and accuracy was evaluated by overall tissue viability for all test substances. The self-assembled RSE exhibited 90% specificity (9/10), 100% sensitivity (0/10), and 95% accuracy (19/20) according to the criteria indicated in the UN GHS Category (Figure 3c). One of the nonirritating chemicals, trans-cinnamaldehyde, had relative tissue viability of less than 50% and was determined inadequate for the criteria, showing overall 90% specificity and 95% accuracy.

These results indicate the excellent performance of the self-assembled RSE regarding the TG439 guideline, as well as its great potential as a toxicity testing tool. Thus, the RSE could be a valuable tool for performing non-animal skin irritation tests. 

### 3.3. Development of an In Vitro Psoriasis-Like RSE Model

To evaluate whether our self-assembled RSE can be used in disease modeling in response to pathogenic cytokines involved in inflammatory skin diseases, we attempted to build a psoriatic skin model using IL-17A, which is a major cytokine in psoriasis pathogenesis. IL-17A was added to the culture medium during the air–liquid interface culture phase of the RSE fabrication procedure. After 2 weeks of culture in the air–liquid interface with IL-17A, the psoriatic RSE model grossly presented a whiter and thicker epidermis than the control model. Histological analysis of the model revealed the presence of psoriatic epidermal characteristics, such as hyperkeratosis, acanthosis, parakeratosis, and hypogranulosis (Figure 4a). Indeed, it was confirmed that the expression of MKi67, an epidermal proliferation marker for Ki67, was increased in the psoriatic RSE model (Figure 4b, Appendix A). Furthermore, in this psoriatic skin model, the expression pattern of keratin subtypes was altered, i.e., KRT6 and KRT17 were upregulated, whereas KRT10 was downregulated (Figure 4c). In turn, psoriasis-associated inflammatory markers, such as IL-1b, hBD2 and S100A9, were significantly upregulated in the psoriatic skin model at both mRNA (Figure 4b and Appendix A) and protein (Figure 4d,e) levels.

Taken together, these results indicate that treatment of IL-17A in our self-assembled RSE model reproduces the macroscopic and histological phenotype of psoriasis and our self-assembled RSE model demonstrates disease-specific characteristics within a 3D structure.

## 4. Discussion

Since the OECD first specified guidelines for predicting the skin irritation potential of medical devices in the 1980s (ISO 10993), additional guidelines, including skin cytotoxicity and hypersensitivity testing methods, were later created [4,10]. Since 2010, the testing guidelines 439 (TG439) have been implemented by several institutes and companies for assessing in vitro test methods for predicting skin irritation potential [7]. In the past, skin irritation and sensitivity tests have been conducted using rabbits or guinea pigs. However, since this causes great discomfort or pain in the animals and there are significant differences in the skin toxicity induced by chemicals depending on the species, in vitro skin irritation test methods have been developed [26]. From this point of view, we verified our self-assembled RSE according to the OECD guidelines and confirmed the disease-modeling potential of this RSE.

A self-assembled RSE model was developed to simulate the full-thickness human skin without the use of exogenous materials, such as murine-, porcine- or bovine-derived ECM [18,19]. The ECM of the self-assembled RSE is composed of human fibroblast-secreted ECM molecules and thus it can be an ideal model for mimicking native skin [27,28]. Our observations revealed that the self-assembled RSE had similar structure and morphology to those of the normal human skin, as assessed using immunohistochemical staining, TEM, and SEM. The self-assembled RSE model has a fully differentiated epidermal layer from the basal layer to the stratum corneum layer, and the self-assembled RSE model demonstrated a well-established basement membrane, including hemidesmosome, lamina densa, lamina lucida, and anchoring fibrils. The self-assembled RSE also expressed barrier proteins and tight junction proteins as in the normal human skin, indicating fully differentiated epidermis with proper barrier function consistent with previous reports [29]. In addition to the morphological and histological similarities to the human skin, the self-assembled RSE showed 100%, 90%, and 95% of sensitivity, specificity, and accuracy, respectively, for predicting irritating chemicals, which fulfilled the OECD criteria. These results suggest that the self-assembled RSE model reproduces the in vivo properties of the human skin, not only in terms morphology but also in terms of epidermal differentiation and barrier function [4,7,8,9,10].

Psoriasis is a chronic inflammatory skin disease that is characterized by well-demarcated erythematous and scaly plaques that can be accompanied by itchiness [30,31] Histologically, in psoriatic lesions, the granular layer of the epidermis is significantly reduced or absent and the mitotic rates of basal keratinocytes are markedly increased, leading to parakeratosis and acanthosis [12]. These characteristics of psoriatic lesions are initiated by proinflammatory cytokines secreted from T cells and are further intensified by forming a positive feedback loop with activated immune cells, such as dendritic cells, neutrophils, and T cells [13]. The pathogenesis of psoriasis includes excessive proinflammatory cytokines, such as IFN-γ, TNF-α, IL-17, IL-22, and IL-23, among which IL-17A plays a key role [14].

In general, the IL-17A cytokine has been used alone or in combination with other cytokines to simulate psoriasis-like inflammation in 2D cell culture systems. In those models, the keratinocytes stimulated with IL-17A exhibited psoriasis-like inflammatory responses [3,32]. As in the 2D culture model, in this study, the IL-17A cytokine was used as a stimulator to induce psoriasis-like skin inflammation. Several 3D skin models of psoriasis have been reported [32,33,34,35,36]. Most of them were constructed using cytokines such as IL-17A, IL-22, IL-1α, IL-6, and TNF-α. Other models incorporated Th1/Th17 polarized CD4^+^ T cells [35] or patient-derived CLA+CCR6+ T cells [36] to reproduce psoriatic phenotype in the 3D skin models. Another model was developed using psoriatic fibroblasts and keratinocytes. This type of model was constructed using either psoriasis-derived fibroblasts or keratinocytes and normal-derived keratinocytes or fibroblasts, or both psoriasis-derived fibroblasts and keratinocytes [12,33,34]. In common, these models of psoriasis showed psoriatic characteristics, such as epidermal hyperproliferation, acanthosis, parakeratosis, and altered expression of psoriatic genes, including *S100A9* and *hBD2*, similar to the results of the present study. Our results suggest that the IL-17A cytokine, which is a critical initiator of the pathogenesis of psoriasis, is sufficient to induce a psoriatic inflammatory response in a 3D skin model.

Because our psoriatic skin model is induced by IL-17A alone, it only recapitulates some part of the psoriasis pathogenesis, not all the aspects. However, it is meaningful that the self-assembled RSE model is very responsive to cytokine stimuli and some of the psoriasis phenotypes can be induced using IL-17A alone in a self-assembled 3D skin model composed of normal-derived fibroblasts and keratinocytes. Considering that blocking IL-17A is a fast and effective therapeutic strategy for psoriasis, it may be possible to exploit the self-assembled RSE model to identify key acting cytokines in disease pathogenesis. Collectively, the self-assembled RSE model of psoriasis is useful in that it can simulate the disease more easily and conveniently and can be applied to construct various skin immune disease models other than psoriasis.

## 5. Conclusions

In this study, we verified that the self-assembled RSE model had a great potential as a testing tool for various pharmaceutical or cosmetic ingredients in the field of skin research. Moreover, this self-assembled RSE was shown to be a worthy means to overcome the limitations of in vitro 2D models, as well as a good alternative to animal experiments. In addition, we successfully reproduced a psoriatic epidermal phenotype using IL-17A, suggesting the potential usage of a self-assembled RSE in modeling and investigating the pathogenesis of inflammatory skin diseases or in drug testing for these diseases.

## Figures and Tables

**Figure 1 pharmaceutics-14-01211-f001:**
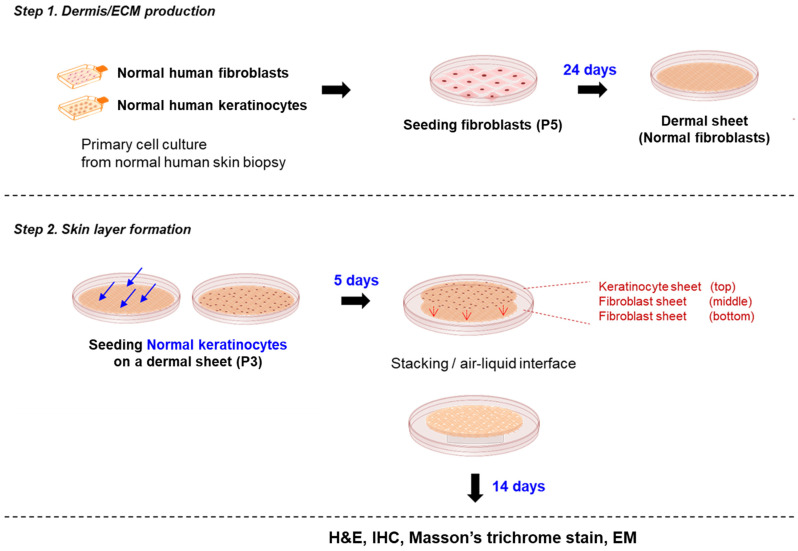
Overview of constructing an in vitro RSE model by the self-assembly tissue engineering method.

**Figure 2 pharmaceutics-14-01211-f002:**
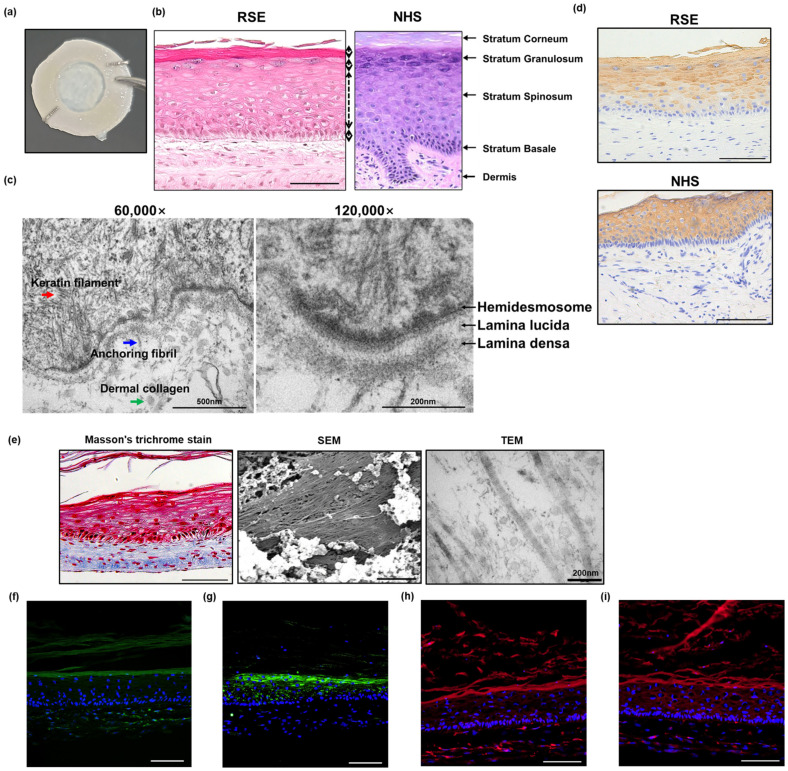
Histological morphology of self-assembled RSE model. (**a**) Macroscopic aspect of the full-thickness skin equivalents produced according to the self-assembly approach of tissue engineering; (**b**) comparison between NHS and fully differentiated self-assembled RSE model by H&E staining (Bar = 100 µm); (**c**) basement membrane structure of the self-assembled RSE model by TEM; (**d**) comparison of KRT10 protein expression between NHS and self-assembled RSE model analyzed by immunohistochemical staining (Bar = 100 µm); (**e**) collagen in the dermis of self-assembled RSE analyzed by Masson’s trichrome stain (Left, Bar = 100 µm), SEM (Middle, Bar = 5 µm), and TEM (Right, Bar = 200 µm); (**f**–**i**) immunofluorescence staining analysis of (**f**) filaggrin, (**g**) involucrin, (**h**) ZO-1, and (**i**) claudin-1 in self-assembled RSE model (Bar = 100 µm, *n* = 4). NHS: Normal human skin; RSE: Reconstructed skin equivalent.

**Figure 3 pharmaceutics-14-01211-f003:**
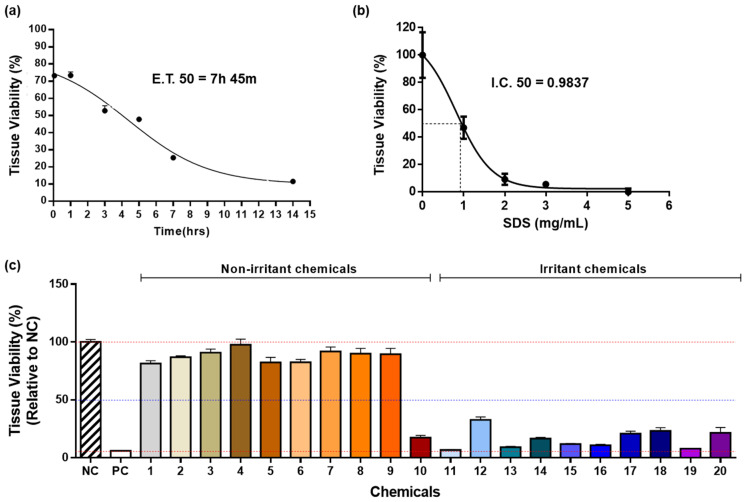
List of chemical-toxicity test substances recommended by OECD guidelines. (**a**) relative viability of self-assembled RSE model according to 1% Triton X-100 treatment time and exposure time required to reduce tissue viability by 50% (*n* = 3); (**b**) relative viability of the self-assembled RSE model according to various SDS concentrations and inhibitory concentration required to reduce tissue viability by 50% (*n* = 3); (**c**) tissue viability in self-assembled RSE models for 20 substances (*n* = 3). Different colors represent each chemical. NC: Negative control; PC: Positive control.

**Figure 4 pharmaceutics-14-01211-f004:**
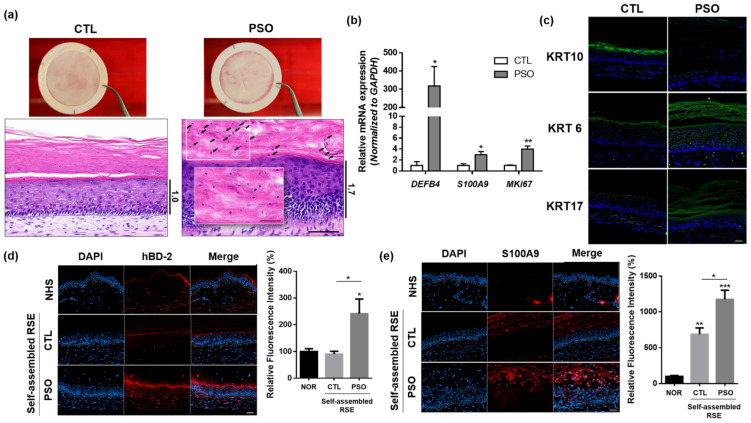
Psoriatic phenotypes of the IL-17A-induced self-assembled RSE models. (**a**) histological analysis of self-assembled psoriasis RSE model (Bar = 100 µm, *n* = 4). The arrows indicate remaining nuclei in the stratum corneum. (**b**) The mRNA expression of *DEFB4, S100A9,* and *MKi67* in the self-assembled RSE model was analyzed by RT-qPCR. (**c**) Immunofluorescence staining analysis of KRT6, KRT10, and KRT17 in self-assembled psoriasis RSE model (Bar = 50 µm, *n* = 4). (**d**) Immunofluorescence staining of HBD-2 in normal human skin and self-assembled RSE (Bar = 50 µm, *n* = 4). For quantification, the fluorescence intensity of the red signals was examined by Image J. (**e**) Immunofluorescence staining of HBD-2 in normal human skin and self-assembled RSE (Bar = 50 µm, *n* = 4). For quantification, the fluorescence intensity of the red signals was examined by Image J. The self-assembled RSE models were treated with vehicle (D.W) for control or 100 ng/mL of IL-17A for the psoriatic model is treated to the culture medium. RSE: Reconstructed skin equivalent; NHS: Normal human skin; PSO: Psoriatic RSE. * *p* < 0.05; ** *p* < 0.01; *** *p* < 0.005.

**Table 1 pharmaceutics-14-01211-t001:** List of chemical-toxicity test substances recommended by OECD guidelines.

Order	Test Chemicals	Formulation	UN GHS Cat.
1	1-Bromochlorobutane-	L	No
2	Diethyl phthalate	L	No
3	1-Naphthalene acetic acid	S	No
4	Propylene glycol	L	No
5	Isopropanol	L	No
6	4-Methyl-thio-benzaldehyde	L	No
7	Methyl stearate	S	No
8	Mineral oil	L	No
9	Hexyl salicylate	L	No
10	*trans*-Cinnamaladehyde	L	No
11	1-decanol	L	R38
12	1-Bromohexane	L	R38
13	5% Potassium hydroxide (5% aq.)	L	R38
14	Di-*n*-propyl disulfide	L	R38
15	2-Benzyloxyethanol	L	R38
16	Heptanal	L	R38
17	Tetrachloroethylene	L	R38
18	Decanoic acid	B	R38
19	Alpha-terpineol	B	R38
20	Butyl Methacrylate	L	R38

## Data Availability

No datasets were generated or analyzed during the current study.

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
