# Peer review of "Structural and Functional Validation of a Full-Thickness Self-Assembled Skin Equivalent for Disease Modeling"

_pharmaceutics, 2022, doi:10.3390/pharmaceutics14061211_

Round 1

Reviewer 1 Report

This manuscript proposed a novel method to develop reconstructed skin using as a tool for in vitro psoriasis disease investigation.  The authors did great work in developing the skin model and validation. However, a few small issues should be addressed or additionally discussed in order to publish.

To investigate the proliferative stage of the dermis layer, the expression level of Ki67 should also be presented.

As this developed reconstructed skin model will be represented human skin barriers, did the authors measure the hydrophobicity of both developed skin models (control and PRO) compared to human skin?

In terms of skin irritation testing, did the author validate the expression level of IL-1B in the developed tissue, which is another essential factor for investigating skin irritation based on OECD 439.

Did the author measure the skin thickness of PSO-derived RSE models compared to normal skin?

The last paragraph of page 4 and the first paragraph of page 5 were redundant.

The quality of all figures was too low. Especially, Figure 4a, yellow arrows were hardly seen.

Author Response

This manuscript proposed a novel method to develop reconstructed skin using as a tool for in vitro psoriasis disease investigation.  The authors did great work in developing the skin model and validation. However, a few small issues should be addressed or additionally discussed in order to publish.

Point 1: To investigate the proliferative stage of the dermis layer, the expression level of Ki67 should also be presented.

Response 1: Thank you very much for your comments on our manuscript. We have conducted the experiments following your suggestion. We have confirmed the mRNA expression level of Ki67 in the control RSE and the PSO-RSE models; the MKi67 mRNA expression was significantly increased in the PSO-RSE model than in the control RSE, and the results are shown in Figure 4b of the revised manuscript.

Point 2: As this developed reconstructed skin model will be represented human skin barriers, did the authors measure the hydrophobicity of both developed skin models (control and PRO) compared to human skin?

Response 2: Thank you for your valuable suggestion. Hydrophobicity is an important ability of the skin in order to perform its protective functions. Unfortunately, we were not able to conduct with your suggestion during this revision period because it takes 8 weeks to construct the RSE model as we mentioned in the method part of our manuscript. However, when we performed the experiments according to the OECD guideline TG439, we observed that distilled water and water-soluble test materials did not absorb into our RHE skin. In the future, it will be interesting to measure the hydrophobicity of our model to represent its proper barrier function.

Point 3: In terms of skin irritation testing, did the author validate the expression level of IL-1B in the developed tissue, which is another essential factor for investigating skin irritation based on OECD 439.

Response 3: Thank you for your advice. We measured IL-1b in our RHE skin and found that it is increased in the PSO-RSE model compared to the control RSE. This result suggests that our RHE model responds to stimuli that alter IL-1b production. It will be interesting to measure IL-1b in response to irritant and non-irritant test materials in the future.

 Point 4: Did the author measure the skin thickness of PSO-derived RSE models compared to normal skin?

Response 4: Thank you for your kind asking. The epidermal thickness of the skin varies from 10 μm to 300 μm depends on the body sites, and our RHE model has a thickness of 170 μm. In addition, our results have shown that the PSO-RSE models have a thicker epidermis (250 μm) than the normal model.

Point 5: The last paragraph of page 4 and the first paragraph of page 5 were redundant.

Response 5: We have revised it.

Point 6: The quality of all figures was too low. Especially, Figure 4a, yellow arrows were hardly seen

Response 6: Thank you for your advice. We have corrected it as per your suggestion.

Reviewer 2 Report

The manuscript by Bo Ram Mok et al., on " Structural and functional validation of a full-thickness self-assembled skin equivalent for disease modeling " is well structured, and the results are well-interpreted but should be improved before publication. 
1.    The authors need to improve the quality of images in the manuscript, especially figures 2 and 4. 
2.    Moreover, SEM image’s in the manuscript are poor, please provide better images, including their scale.
3.    I would like to ask the authors if they have measured the transepidermal water loss (TEWL) for the RSE model? If not, please provide the value.
4.    Please, add the conclusion section in the manuscript.

Author Response

The manuscript by Bo Ram Mok et al., on " Structural and functional validation of a full-thickness self-assembled skin equivalent for disease modeling " is well structured, and the results are well-interpreted but should be improved before publication. 

Point 1: The authors need to improve the quality of images in the manuscript, especially figures 2 and 4.

Response 1: Thank you very much for your comments on our manuscript. We have improved the quality of images following your suggestion.

Point 2: Moreover, SEM image’s in the manuscript are poor, please provide better images, including their scale.

Response 2: We have changed the image following your advice.

Point 3: I would like to ask the authors if they have measured the transepidermal water loss (TEWL) for the RSE model? If not, please provide the value.

Response 3: Thank you for your kind suggestion. It would be great to measure TEWL or prove its value with our RHE model. However, our 3D RHE model is an in vitro model cultured at 37C in a 5% humidified CO2 atmosphere with culture media, and for the measurement of TEWL in human skin, a room of temperature 18–21 °C and relative humidity of 40%–60% should be used. In addition, different in vitro models give different TEWL measurements. Therefore, the value of TEWL from the RHE model cannot be compared with it from human skin. Instead of measurement of TEWL, we showed proper granular layer and cornified layer, and the appropriate expression of barrier proteins and tight junction proteins. In the future, measurement of TEWL would be a great help to validate the atopic dermatitis model where skin barrier dysfunction is its key pathogenesis.

Point 4:  Please, add the conclusion section in the manuscript.

Response 4: Thank you for your suggestion. We separated the content of the conclusion part from the discussion section and presented it in the conclusion section of the revised manuscript.

Round 2

Reviewer 1 Report

Thank you for your response.

Please put the data of IL-1B expression and skin thickness of RSE and PSO-RSE model as supplementary data. 

Author Response

Point 1: Please put the data of IL-1b expression and skin thickness of RSE and PSO-RSE model as supplementary data.

Response 1: Thank you very much for your comments on our manuscript. As per your suggestion, we have reorganized the data into supplementary data.
